# Biofilm Development on Carbon Steel by Iron Reducing Bacterium *Shewanella putrefaciens* and Their Role in Corrosion

Sachie Welikala [1], Saad Al-Saadi [1,2], Will P. Gates [3], Christopher Panter [4] and R. K. Singh Raman [1,2,*]

1 Department of Chemical and Biological Engineering, Monash University, Clayton, VIC 3800, Australia; swelikala@hotmail.com (S.W.); saad.al-saadi@monash.edu (S.A.-S.)
2 Department of Mechanical and Aerospace Engineering, Monash University, Clayton, VIC 3800, Australia
3 Melbourne-Burwood Campus, Institute for Frontier Materials, Deakin University, Burwood, VIC 3125, Australia; will.gates@deakin.edu.au
4 CP Microbiology and Analytical Laboratories, 677 Springvale Rd., Mulgrave, VIC 3170, Australia; services@cpmicro.com.au
* Correspondence: raman.singh@monash.edu

**Abstract:** Microscopic, electrochemical and surface characterization techniques were used to investigate the effects of iron reducing bacteria (IRB) biofilm on carbon steel corrosion for 72 and 168 h under batch conditions. The organic nutrient availability for the bacteria was varied to evaluate biofilms formed under nutritionally rich, as compared to nutritionally deficient, conditions. Focused ion beam-scanning electron microscopy (FIB-SEM) was used to investigate the effect of subsurface biofilm structures on the corrosion characteristics of carbon steel. Hydrated biofilms produced by IRB were observed under environmental scanning electron microscope (ESEM) with minimal surface preparation, and the elemental composition of the biofilms was investigated using energy dispersive spectroscopy (EDX). Attenuated total reflectance-Fourier transform infrared spectroscopy (ATR-FTIR) was used to provide information on the organic and inorganic chemical makeup of the biofilms. Electrochemical techniques employed for assessing corrosion, by open circuit potential, linear polarization and potentiodynamic polarization tests indicated no significant difference in the corrosion resistance for carbon steel in IRB-inoculated, compared to the abiotic solutions of common Postgate C after 72 and 168 h. However, the steel was found to be more susceptible to corrosion when the yeast extract was removed from the biotic environment for the 168 h test. In the absence of yeast nutrient, it is postulated that IRB received energy by transforming the protective film of $Fe^{3+}$ into more soluble $Fe^{2+}$ products.

**Keywords:** Iron reducing bacteria (IRB); focused ion beam-scanning electron microscopy (FIB-SEM); environmental scanning electron microscopy (ESEM); microbiologically influenced corrosion



## 1. Introduction

Microbiologically influenced corrosion (MIC) is the term designated to the acceleration of a corrosion process due to the activities of microorganisms where the electrochemical reactions are enhanced because of bacterial metabolic products [1–4]. Microorganisms contribute to corrosion processes by either direct influence on the rate of anodic or cathodic reactions or by changing surface film characteristics due to biofilm formation [4]. The structural and chemical heterogeneities of the biofilm most commonly causes localized corrosion [5,6]. Moreover, due to the influence of microorganisms and their metabolic processes, corrosion may be observed in unlikely conditions such as anoxic [7] or low chloride environments where corrosion would otherwise not occur [8].

Iron reducing bacteria (IRB) derive energy from the reduction of ferric ions ($Fe^{3+}$) using either the electron transport chain in anaerobic respiration or via fermentation [9]. Influence of IRB on corrosion is a less explored topic, and the findings in the reported studies are conflicting, with some observing corrosion protection and others showing

corrosion acceleration in presence of IRB [10–12]. Such inconsistencies are due to difference in species of IRB or the composition of electrolyte used in laboratory experiments [10]. In the limited studies on the topic, the corrosion acceleration due to IRB is attributed to the bacteria's ability to produce soluble ferrous irons through reductive dissolution of the protective corrosion film of insoluble ferric iron oxides on the steel surface, thereby exposing the alloy surface to aggressive ions in the environment [10,12,13]. However, a few laboratory studies have reported the protective effect of IRB [14,15]. Lee and Newman reported that even if biofilm coverage of the metal is incomplete, cumulative $O_2$ respiration from both planktonic and sessile cells rapidly depletes $O_2$ on the metal surface, thus inhibiting the galvanic process and corrosion [16]. In fact, Dubiel et al., reported that mutant S. oneidensis sp. could be used as a potential method to control corrosion in pipeline systems as the bacteria may colonise the metal surface and aerobically consume oxygen molecules adjacent to the metal surface [15]. As oxygen is depleted, the facultative anaerobic IRB turn to $Fe^{3+}$ reduction and produce $Fe^{2+}$ ions which diffuse into the bulk fluid. This process has been reported to create a chemical shield that further reduces oxygen diffusion, which in turn may inhibit the cathodic reaction due to lower oxygen availability. By electrochemical reaction, $Fe^{2+}$ is oxidized to $Fe^{3+}$ and is available again for reduction by bacterial respiration [17,18].

IRB Shewanella genera are relevant to corrosion as they are among the most common bacteria in oilfield injection water [19]. The bacteria has the ability to reductively dissolve iron oxyhydroxide minerals (that are generally insoluble at neutral pH [20–23]) into soluble ferrous ion or secondary mineralised Fe(II) oxides such as magnetite ($Fe_3O_4$), siderite ($FeCO_3$), vivianite ($Fe(PO_4)_2$) and green rusts [21,24–26].

The objective of this work is to characterise the IRB biofilm on carbon steel, and its effect on corrosion characteristics. In particular, the biofilms were manipulated by varying the organic carbon nutrients available to the bacteria so that the biofilms developed under: (i) high organic nutritional conditions such as the commonly used Postgate C medium; (ii) where organic nutrients were moderately decreased by the removal of chemically un-defined nutrient yeast extract (modified Postgate C medium) and (iii) where the biofilm formed under starvation conditions, where all organic nutrients were removed (i.e., the inorganic medium). Biofilms formed by a representative IRB (*Shewanella putrefaciens*) were investigated using microscopic observation, chemical characterisation and electrochemical techniques. Whether biofilms accelerate or inhibit corrosion processes on carbon steel is highly dependent on environmental conditions available to the bacteria or by the ability of the bacteria to alter the environment. The biofilms formed on carbon steel were investigated for 72 h and 168 h in pure culture of IRB. Open circuit potential (OCP), linear polarization (LP) and potentiodynamic polarization (PDP) tests were used for characterization of corrosion. Environmental scanning electron microscopy (ESEM) was used to investigate the morphologies of the biofilm on the carbon steel specimen. To the best of authors' knowledge this is among the first comprehensive characterisation of cross-sections of biofilms and MIC using focused ion beam-scanning electron microscopy (FIB-SEM) that enabled investigation of the sub-surface features of the biofilm, particularly at the locations of specific surface features recognised by SEM. Attenuated Fourier transformation infrared spectroscopy (ATR-FTIR) was used, alongside microscopic and electrochemical techniques, to observe the extracellular polymeric substances (EPS) produced by IRB.

## 2. Materials and Methods

### 2.1. Maintenance of Bacterial Culture

Pure bacterial culture of IRB (*S. putrefaciens*) was grown in test tubes under aerobic conditions in an amended B7 medium at a mesophilic temperature of 30 °C. The composition of the amended B7 medium (henceforth referred to as B7) was (/L): 0.8 g $K_2HPO_4$, 0.2 g $KH_2PO_4$, 0.2 g $MgSO_4$, 5 g yeast extract, 5 g peptone, 10 g ferric ammonium citrate, 10 mL $CaSO_4$ (saturated solution). The pH was adjusted to 7.5–7.8 using NaOH solution. All lab glassware pipettes and solutions were autoclaved at 121 °C for 15 min to avoid

contamination of the enrichment cultures. The cultures were periodically renewed to keep them active and healthy. A 1 mL sample of a three-day old inoculum was taken from the culture for biofilm development and electrochemical experiments.

### 2.2. Test Solutions Preparation

Three recipes of Postgate C medium were prepared for biofilm development and corrosion characterization (i.e., commonly used Postgate C, modified Postgate C and inorganic Postgate C solutions). Table 1 shows the composition of the three Postgate C used in the present study. The modified Postgate C medium was prepared by removing the yeast extract from the medium to moderately decrease the amount of organic nutrients available to the bacteria. The inorganic medium was prepared by completely removing all organic ingredients in the medium in order to study the behaviour of the bacteria under starvation conditions. The pH of the culture medium was adjusted to 7.5–7.8 using sterilised NaOH solution and all solutions were autoclaved to avoid contamination. No iron component was included in the test media for electrochemical testing or for biofilm study. The iron in solution that originated from the dissolution of the carbon steel specimen was measured from the 1, 10 phenanthroline method.

**Table 1.** Chemical composition of the commonly used Postgate C [26], modified Postgate C and inorganic Postgate C solutions.

| Chemicals | Postgate C | Modified Postgate C | Inorganic Postgate C |
|---|---|---|---|
| $Na_2SO_4$ [a] | 4.5 | 4.5 | 4.5 |
| $CaCl_2 \cdot 2H_2O$ [a] | 0.06 | 0.06 | 0.06 |
| Lactic acid [b] | 4.8 | 4.8 | - |
| Sodium citrate [a] | 0.3 | 0.3 | - |
| $NH_4Cl$ [a] | 1.0 | 1.0 | 1.0 |
| $K_2HPO_4$ [a] | 0.5 | 0.5 | 0.5 |
| $MgSO_4 \cdot 7H_2O$ [a] | 2.0 | 2.0 | 2.0 |
| Yeast extract [a] | 1.0 | - | - |

[a] Concentration in g/L. [b] Concentration in mL/L.

### 2.3. Carbon Steel Sample Preparation

For electrochemical test, the preparation of steel specimens is described in detail in [27–29]. A copper wire was soldered to a square specimen of the carbon steel. The wire was encased in a glass tube to prevent any contact of the wire with the electrolyte. Each specimen was then mounted in epoxy resin so that only one face of the sample was exposed to the test solution. The surface was wet ground with emery paper up to 1200 grit and then polished with diamond paste up to 3 μm. The edges of the resin and the metal were painted with enamel coating to avoid crevice corrosion [27–30]. For samples used for microscopic investigation, a fishing line was used to hang the sample in the test solution. The samples were ultrasonically cleaned for 15 min in ethanol prior to being used as test electrodes in electrochemical experiments or in immersion tests for biofilm characterization.

### 2.4. Iron Quantification

Due to the importance of ferrous and ferric ion concentrations on the corrosion characteristics of carbon steel, the total iron ($Fe^{2+}$ and $Fe^{3+}$) concentration in this study was determined using a colorimetric technique using 1, 10-phenanthroline according to the procedure outlined in reference [31].

### 2.5. Microscopic and Biofilm Cross-Sectional Analysis

2.5.1. Development of Biofilms on Carbon Steel for Microscopic Imaging and FIB-SEM Analysis

The carbon steel specimens, mounted in epoxy resin (as described in Section 2.3) were attached to a polypropylene fishing line, and immersed in 250 mL of sterilized test solutions in a 250 mL Duran bottle. Using aseptic techniques, 1 mL of bacterial culture

was taken from a 3-day old stock culture of IRB and inoculated into the Duran bottle as pure culture. The solution was not de-aerated. An abiotic control was also set up under the same conditions. The carbon steel specimen was exposed to the bacterial culture (or abiotic conditions) for a period of 72 h or 168 h at 30 °C. At least two samples were observed for each condition in order to examine reproducibility.

### 2.5.2. Biofilm Fixation and Environmental Scanning Electron Microscopy (ESEM) Imaging

The morphology of the biofilm on the carbon steel specimen was observed using the 3D Quanta FEI in ESEM mode. Biofilms were fixed using 2% glutaraldehyde solution for 1 h and then subject to two washes in deionised water, each wash for 5 min. The use of relatively high pressures (~650 Pa, or 1 mbar) and water vapor in the specimen chamber in the ESEM mode allowed the samples to be viewed in a hydrated state, and without the need for metallic coating, as is often required in SEM sample preparation which introduces artifacts. Minimum of two samples were observed under ESEM for each condition.

### 2.5.3. Focused Ion Beam—Scanning Electron Microscopy (FIB-SEM)

FIB-SEM technique was used to investigate sub-surface features of the biofilm. FIB milling was carried out using the FEI Quanta 3D FEG instrument. As this instrument also has SEM and EDS capabilities, the subsurface structure could be viewed, and elemental composition analysed following the milling. Due to the sample being placed in a high vacuum chamber (~$10^{-6}$ Pa), in addition to the sample preparation steps detailed in Section 2.5.2 for ESEM, the sample was also dehydrated and sputter coated with 3 nm of platinum coating in order to avoid charging. The surface of the specimen was first viewed under SEM to locate a site of interest in the biofilm for subsequent cross-sectional analysis. The site chosen for FIB milling was typical and representative for the condition under study. A platinum strip, 1 µm in width and 1 µm in thickness, was applied across the length of the area of interest in order to protect the biofilm from ion beam degradation during the milling process [32]. The sample was then tilted to 52° so that the milling could be performed using current gallium (Ga$^+$) beam (in the range of 5 nA). Beam currents were lowered (in the pA range) for subsequent cleaning of cross-sections to remove material re-deposited on the area during the initial rough milling.

### 2.5.4. Attenuated Total Reflectance—Fourier Transform Infrared (ATR-FTIR) Spectroscopy

The specimens exposed to biotic and abiotic conditions were removed after 72 h and any loosely attached bacterial cells were rinsed off with phosphate buffer solution. The specimens were left in a desiccator for at least 24 h to dehydrate. FTIR spectra were obtained using a Perkin Elmer Spectrum 100 series spectrometer. The spectral acquisition (128 coadded scans at 8 cm$^{-1}$ of spectral resolution in the 4000 cm$^{-1}$ to 600 cm$^{-1}$ range).

### 2.6. Electrochemical Measurements

Linear polarization (LP) and potentiodynamic polarization (PDP) were carried out using a Princeton Applied Research Potentiostat (model: 2273) and a three-electrode electrochemical cell. The test cell was a 250 mL Duran bottle filled with 250 mL of the test solution. The design of the electrochemical cell is described in [27–29]. The steel specimens were used as the working electrodes, saturated calomel electrode (SCE) was used as the reference electrode and a platinum mesh was used as the counter electrode. For the biotic electrochemical experiments, 1 mL of inoculum from the maintenance culture was introduced into the test solution. Abiotic control experiments were also conducted alongside the biotic experiments. All the measurements were repeated at least thrice to examine reproducibility. Open circuit potential (OCP) was monitored to confirm its stability with time. For LP test, the working electrode was polarized ±10 mV from E$_{corr}$ at a scan rate of 0.166 mV/s. Potentiodynamic polarization was carried out at a scan rate of 1 mV/s, starting at a potential of 400 mV more negative to the OCP.

## 3. Results and Discussion

### 3.1. Iron and Sulfide Measurements in Solution

The B7 medium was used as the enrichment medium for maintaining the IRB culture. This medium is designed especially for the IRB [33], and therefore, contains high contents of ferric ammonium citrate (10 g/L) and yeast extract (5.0 g/L). Figure 1 follows the growth of the IRB in the B7 medium over a 48-day period. Initial colour, at the time of IRB inoculation (at day 0) is brown. This colour gradually changes to dark green, which indicates the ability of IRB to reduce $Fe^{3+}$ ions from ferric ammonium citrate to the ferrous ions ($Fe^{2+}$) [33]. From day 3 onwards, a gelatinous substance was observed to have deposited at the bottom of the test tube (though not clearly visible in Figure 1). This precipitate was sticky and difficult to remove from the test tubes, at the end of experiments.

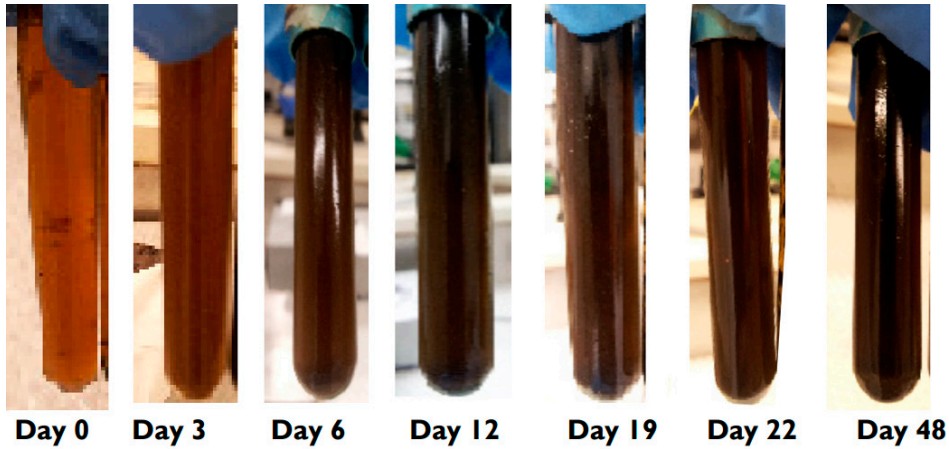

**Day 0    Day 3    Day 6    Day 12    Day 19    Day 22    Day 48**

**Figure 1.** Growth of the IRB, *Shewanella putrefaciens* in B7 medium.

Dissolved sulfide ions were not detected in the solution in the B7 or Postgate C medium containing IRB when tested spectrometrically, using the $CuSO_4$ technique. This confirmed that the bacterium used in this particular study was unable to reduce sulfate, as also reported in the literature [34].

Figure 2 shows quantification of total iron ($Fe^{2+}$ and $Fe^{3+}$) in the modified Postgate C medium in the presence of the IRB and carbon steel specimen. As suggested, the concentration of iron increased over time in the presence of the IRB, form 2.2 mg/L after 72 h to 21 mg/L after 168 h. The IRB is known to reduce insoluble Fe(III) oxides such as goethite, hematite, ferrihydrite, akaganéite, and lepidocrocite into soluble ferrous complexes [12,21,35] and this is the most likely reason for the much higher dissolved iron concentration in biotic solution containing IRB.

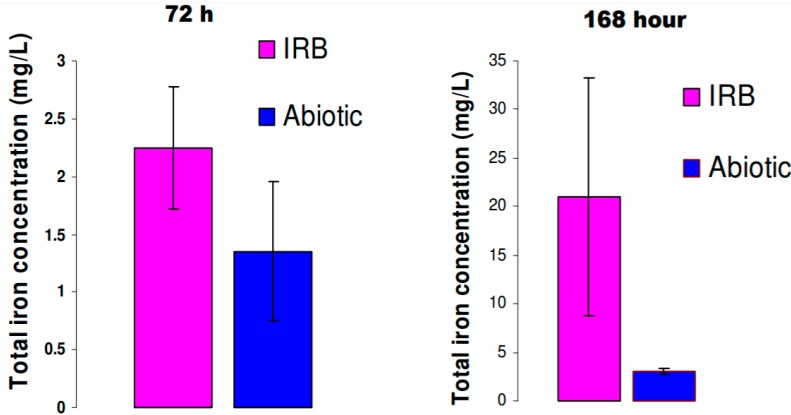

**Figure 2.** Quantification of total iron ($Fe^{2+}$ and $Fe^{3+}$) in solution after 72 and 168 h in modified Postgate C medium in the presence of the IRB and carbon steel specimen.

### 3.2. Biofilm Development and Electrochemical Characterization of Carbon Steel in Biotic Postgate C Solution

3.2.1. Biofilm Development in Commonly Used Postgate C Solution with IRB

Following 72 h exposure of carbon steel to the IRB in the Postgate C medium, the specimen was covered by EPS-like material (Figure 3). Figure 3a shows a globular deposit of ~2 μm diameter. Much smaller globular precipitates can be observed in Figure 3b which are packed close together. Pits covered by loose corrosion product cap, as shown in Figure 3c,d, were also a common occurrence throughout the sample surface.

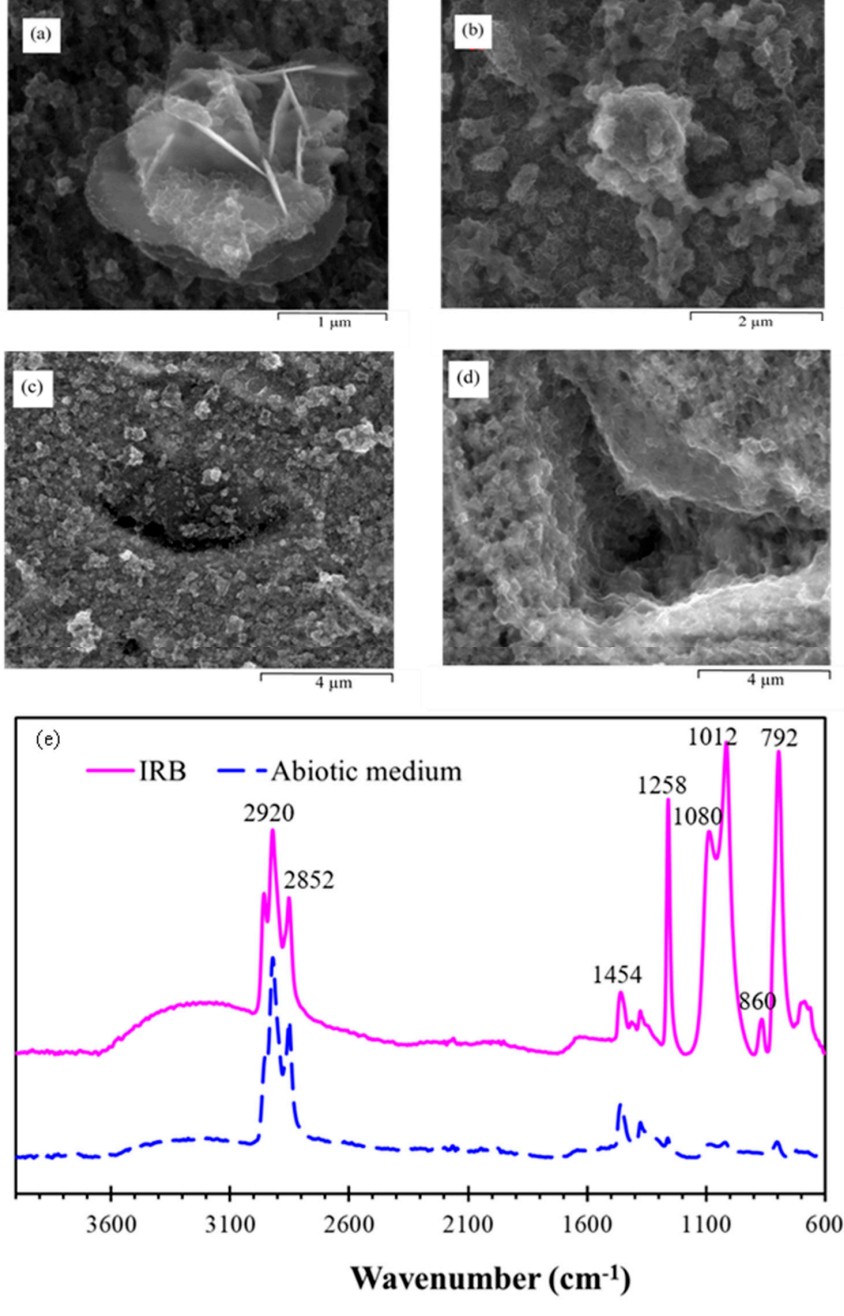

**Figure 3.** ESEM images of hydrated biofilm formed on carbon steel by a pure IRB culture following 72 h exposure to Postgate C medium, (**a**,**b**) reduced Fe(II) oxide species, (**c**) sites of pitting attack under loose cap of corrosion product and (**d**) pit initiated at a MnS inclusion site. (**e**) IR spectra of IRB biofilm formed on carbon steel upon to exposure to pure IRB culture in Postgate C medium and abiotic Postgate C for a period of 72 h.

The IRB is capable of reducing minerals such as amorphous hydrous ferric oxide (HFO), goethite, lepidocrocite or ferrihydrite into mixed Fe(III)/Fe(II) species [25]. Little et al. [12] also observed similar globular precipitates when *S. putrefaciens* was grown on top of synthetic goethite or ferrihydrite minerals. EDS carried out on these globular particles following 190 h exposure showed that the composition contained iron, phosphorous and magnesium, suggesting that these particles were residual goethite particles [12]. In the current study, EDS could not be used to distinguish between the chemical compositions of different particles in the 72 h IRB biofilm due to the thickness of the film being less than the interaction volume of the EDS beam. Framboidal structures of Fe minerals are very much associated with the biogenic formation. It has to do with the external surface to total volume ratio of a sphere being the most stable condition thermodynamically.

Uncovered, irregular shaped pits (Figure 3d) were also observed and are likely to be initiated due to dissolution of inclusions. The site of the inclusion at the centre of the pit appears to have completely dissolved and pitting has progressed to a greater extent than the micro-pits observed in the abiotic control. ESEM analysis could not confirm the presence or absence of the IRB bacterial cells on the biofilm. Individual bacterial cells are difficult to image using ESEM as bacteria are mostly composed of water and low atomic number elements which are not very electron dense. It is also likely that the IRB cells are unrecognizable due to being encrusted in the iron oxides and/or biofilm. Direct contact between the IRB and the Fe(III) oxide, however, may not always be necessary, as numerous geomicrobiological studies have reported on the ability of *S. putrefaciens* to reduce Fe(III) from a distance without direct contact with the Fe(III) under anaerobic conditions [36,37]. Lies et al. [37] observed that S. oneidensis reduced at least 86.5% of iron from an "iron-bead" system in the absence of direct contact when under nutritionally rich conditions. This iron reduction from a distance is thought to be carried out through ferric iron chelators or extracellular electron shuttles [36–38]. This is not, however, in agreement with all studies, as some authors have reported that iron reduction did not occur when direct contact between the bacterial cells and the iron oxyhydroxides was impeded [10,12].

Figure 3e depicts IR spectrum of IRB biofilm developed on carbon steel exposed to pure culture of IRB in Postgate C medium for a period of 72 h, with the spectrum of abiotic Poctgate C for comparison. Hydrogen and fatty acids are primary products of IRB metabolism [39] and the peaks corresponding to these fatty acids can be clearly identified for the IRB biofilm. Characteristic peaks corresponding to -$CH_3$ asymmetric stretching (2956 $cm^{-1}$), asymmetric -$CH_2$ stretching (2920 $cm^{-1}$) and symmetric -$CH_2$ stretching (2852 $cm^{-1}$) were observed confirming the presence of organic hydrocarbon chains in the IRB biofilm. Complementary bands for these hydrocarbons were found at 1454 $cm^{-1}$ (asymmetric -$CH_3$ bending/$CH_2$ scissoring) as well as -$CH_3$ symmetric bending at 1372 $cm^{-1}$.

The broad peak at 3200 $cm^{-1}$ is considered to be due to -OH stretch of Fe(III) oxyhydroxides, which are the primary inorganic corrosion product of steel. IRB dissolves Fe(III) oxyhydroxide minerals into Fe(II) oxides as part of anaerobic respiration. These OH groups detected in the IR spectra are attributed to either stoichiometric -OH in iron oxyhydroxide species such as goethite, lepidocrocite, ferrihydrite, but could also be surface bound water present in the iron hydroxide minerals [40]. The broad peak detected at 3200 $cm^{-1}$ in this study could also be due to the formation of the mixed ferric/ferrous species, green rust, due to the bacteria's incomplete reduction of ferric oxyhydroxide and oxide minerals as also observed in other studies in the presence of *S. putrefaciens* [21,41]. Magnetite may also form in the solution if deficient in $Fe^{2+}$ ions as this mineral has smaller Fe(II)/Fe(III) ratio [25,41]. The green rusts precipitated due to IRB bio-mineralization may abiotically re-mineralize into black magnetite precipitates with time [24].

The most intense peaks detected in the IR spectra for the IRB biofilm correspond to the stretching of the phosphate ion. High intensity peaks for asymmetric $PO^{2-}$ stretch at 1258 $cm^{-1}$ and symmetric $PO^{2-}$ stretching at 1080 $cm^{-1}$ were observed. The P-O asymmetric stretch at 860 $cm^{-1}$ was also detected as well as a high intensity peak for P-O

symmetric stretch at 792 cm$^{-1}$. A high intensity vibration at 1012 cm$^{-1}$ was observed in the saccharide region which may correspond to C-O-C, C-O-P, P-O-P ring vibrations of carbohydrates [42–44]. Specific identification is difficult in the saccharide region due to the complex superpositions of the characteristic absorptions of various polysaccharides with the various organophosphates present. The cell wall of gram-negative bacteria, such as *S. putrefaciens*, consists of an outer membrane containing lipopolysaccharides (LPS) [25,45] as well as a periplasm with a thin peptidoglycan layer [45]. Therefore, the C-O-P and P-O-P vibrations may be due to polysaccharides of cell wall. LPS are known to be highly anionic and would therefore contribute to binding of cations such as Fe$^{2+}$ ions from the carbon steel due to electrostatic interactions [24]. Omoike and Chorover [46] have reported that phosphodiester groups of nucleic acids in the EPS may also facilitate the binding of the EPS onto iron oxyhydroxide minerals.

The FIB-SEM cross-sections (Figure 4a,b) show two separate pits covered by a loose corrosion product crust such as that shown in Figure 3c. The depth of these pits was approximately 1.5 μm. The biofilm covering the pit was loosely attached to the corroded carbon steel. The innermost layer was composed of initial layer of colloidal corrosion products. The loose attachment between the metal and biofilm material covering the pit allows chemical species from electrolyte to come in contact with the metal to facilitate pit propagation. High organic material in the IRB biofilm caused charging and sample movement when FIB milling was being carried out. This difficulty in obtaining clean cross-sections from FIB milling only occurred in the IRB pure culture biofilm, due to the higher organic EPS material in the biofilm which were not electron dense.

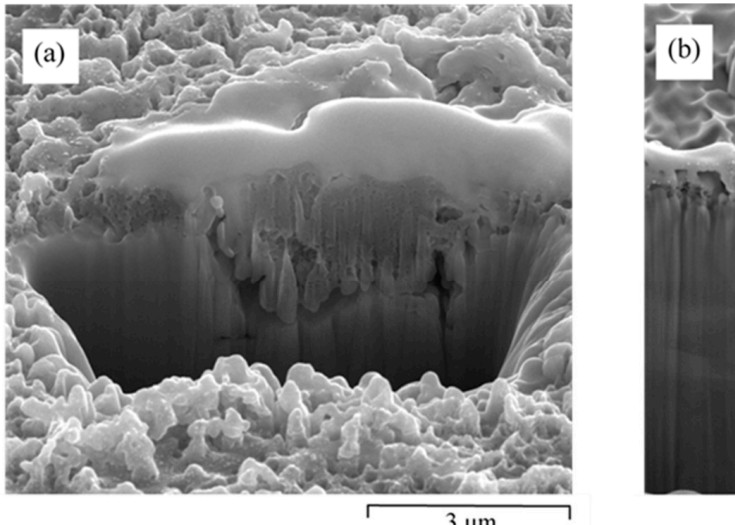
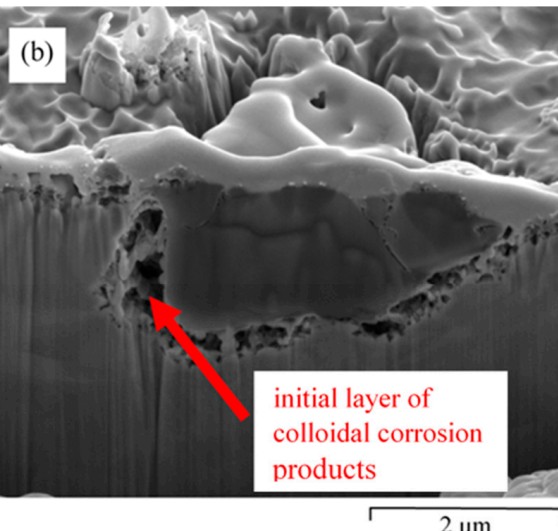

**Figure 4.** (**a**,**b**) FIB-SEM cross-sections of the biofilm formed on carbon steel and corrosion pitting due to pure culture of IRB in Postgate C medium following a 72 h exposure.

Figure 5a,b show ESEM and SEM images of biofilm formed on carbon steel exposed for 168 h to Postgate C medium inoculated with pure IRB culture. Polishing marks were visible on some areas of the carbon steel surface, indicating a low corrosion rate in considerable parts of the specimen (Figure 5a). The surface was covered with a thin and porous biofilm, with corrosion products scattered throughout the sample when observed under ESEM (Figure 5b). EDS analysis of the biofilm corresponding to the ESEM image showed iron and carbon to be the main elements detected in the biofilm as shown in Figure 5c. The detection of carbon is most likely to be due to the EPS material, which is high in hydrocarbons from fatty acids. Alongside the flat areas observed in ESEM analysis, thicker corrosion deposits also occurred throughout the sample. One such deposit was chosen for the FIB-SEM cross-sectional characterization (Figure 5d,e). At high magnification, the thin biofilm

beside the deposit showed initial signs of undercutting attack as the metal was dissolved from underneath.

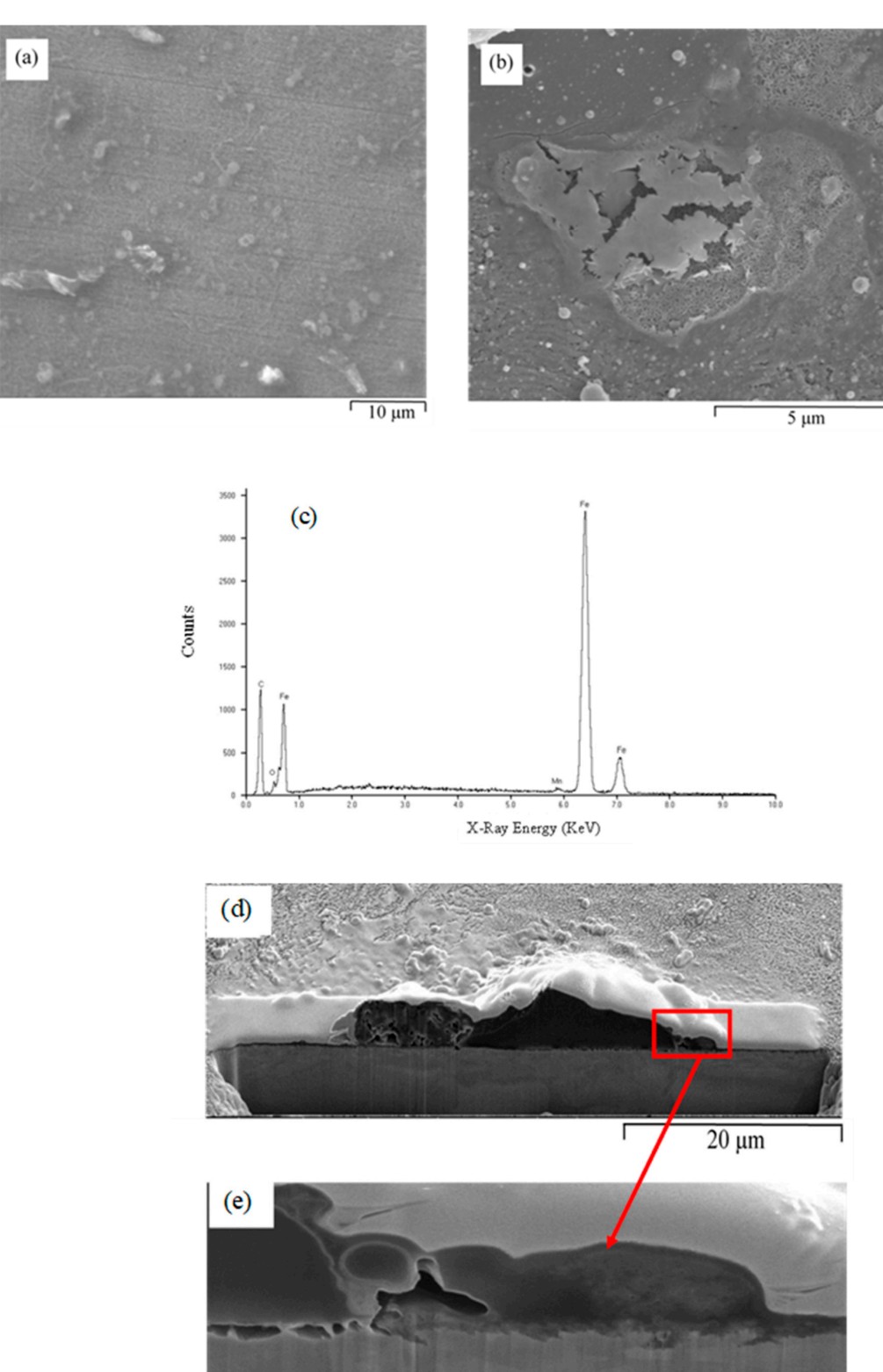

**Figure 5.** (**a**) ESEM and (**b**) SEM images of biofilm formed on carbon steel by a pure IRB culture (168 h exposure time to Postgate C medium). (**c**) EDS analysis of the biofilm corresponding to the ESEM image. (**d**) FIB-SEM images of the cross-section and (**e**) initial layer of fibrous EPS material produced by IRB.

### 3.2.2. Electrochemical Behaviour of Carbon STEEL in Postgate C Medium Inoculated with IRB

Figure 6a shows the polarization resistance ($R_p$) of mild steel exposed during the duration of exposure of 72 h to the Postgate C medium inoculated with IRB culture in comparison to that exposed to the abiotic Postgate C solution. $R_p$ increased from 890 $\Omega/cm^2$ at 0 h to be 7676 $\Omega/cm^2$ at 24 h, presumably due to the removal of oxygen by the IRB as IRB respiration removes oxygen, i.e., an expected cathodic reactant for steel corrosion. From 24 h to 72 h, the $R_p$ remained steady. The $R_p$ of the carbon steel exposed to the IRB had a slightly lower $R_p$ value from around 24 h to 72 h reading than the $R_p$ for abiotic control. This indicated that the IRB biofilm may have somewhat decreased the corrosion protection offered by the abiotic corrosion products, by dissolving iron oxyhydroxides and/or forming porous EPS and reduced Fe(II) oxide species.

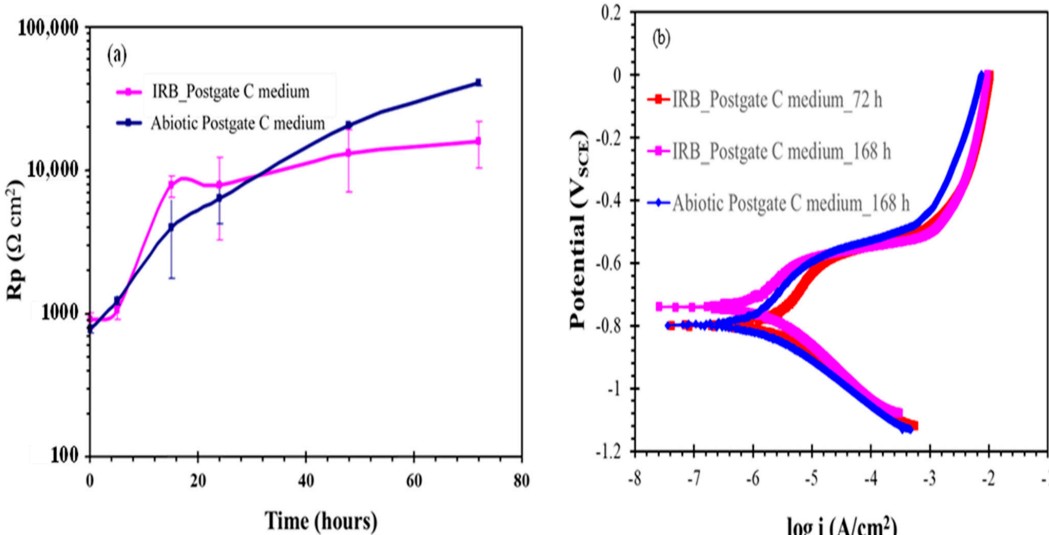

**Figure 6.** (**a**) Polarization resistance (**a**) of carbon steel exposed to a pure IRB culture and abiotic conditions in Postgate C medium, and (**b**) a potentiodynamic polarization and (**b**) scan of carbon steel after 72 h exposure with and without IRB.

Figure 6b shows potentiodynamic polarization curves of the steel samples pre-exposed to the Postgate C solution with IRB for 72 and 168 h. The polarization curve of the steel sample pre-exposed to the abiotic solution for 168 h is shown for comparison. There was no significant difference in the anodic or cathodic regime of the carbon steel exposed to IRB for 72 h or 168 h or to abiotic condition for 168 h. The $E_{corr}$ of the steel samples exposed to the biotic solution is shifted about 50 mV towards positive direction. In all cases, $E_{pit}$ appeared to be around $-550$ mV, where a sudden and rapid increase in anodic current was observed. The IRB biofilm consisting of Fe(II)/Fe(III) oxides and EPS did not offer any resistance to anodic dissolution.

### 3.3. Effect of Removal of Yeast Extract from Postgate C (Modified Postgate C) on IRB Activity and Biofilm Development

#### 3.3.1. Biofilm Development in Modified Postgate C Solution with IRB

Exclusion of yeast extract from the culture medium in the modified Postgate C medium resulted in less coverage of the steel specimen with EPS and biominerals. However, as will be evident from subsequent description, the carbon steel surface exposed to the modified Postgate C medium for 72 h confirmed that IRB metabolized and reduced $Fe^{3+}$ ions even in the absence of the yeast extract. Lovely et al. [39] have also reported that yeast extract was not essential for the growth and anaerobic metabolism of *S. putrefaciens*.

Figure 7 presents ESEM analysis of hydrated biofilm formed on carbon steel by a pure IRB culture, following 72 h exposure in modified Postgate C medium. Most parts of the

carbon steel surface had a very flat, uniformly corroded appearance (Figure 7a). However, as seen in Figure 7b, a higher magnification reveals the hollow iron oxyhydroxide shell. Lepidocrocite crystals in Figure 7c appear to be a mixture of crystalline and amorphous or poorly crystalline material. O'Loughlin has observed through time-resolved XRD that lepidocrocite was reduced to green rust by *S. putrefaciens* [21]. Green rust is a mixed ferrous/ferric hydroxide which has a structure consisting of alternating layers of positively charged hydroxide and hydrated anion often containing carbonate [21]. The morphology of the green rust crystal is reported to vary with environmental conditions. The green rust crystals are hexagonal in the presence of $SO_4^{2-}$ alone [47,48] but rhombohedral in solutions containing $Cl^-$, $SO_4^{2-}$ or $Br^-$ [47]. Globular aggregates, similar to that shown in Figure 7d, have been assigned to be siderite in study by Fredrickson et al. [25]. Due to the thin nature of the film, similar EDS spectra were obtained for all minerals across the biofilm. Carbon, oxygen, iron, phosphorous were the common elements detected through EDS (Figure 7e), which suggested that most minerals observed were different morphologies of the same phases (mainly iron oxyhydroxides). Elemental sulphur was also detected in some areas of the film.

The FTIR spectra for the biofilm in the modified Postgate C at 72 h (Figure 7f) was generally similar to that observed in the Postgate C medium (Figure 3e). Due to the low level of organic nutrients available for the bacteria, lower amounts of EPS were produced by the IRB, as suggested by the weaker absorbance signal in the modified Postgate C medium. The presence of hydrocarbon chains of lipids was confirmed by the characteristic peaks corresponding to $-CH_3$ asymmetric stretching (2958 $cm^{-1}$), asymmetric $-CH_2$ stretching (2916 $cm^{-1}$) and symmetric $-CH_2$ stretching (2848 $cm^{-1}$). Complementary bands for these hydrocarbons were found at 1452 $cm^{-1}$ (asymmetric $-CH_3$ bending/$CH_2$ scissoring), $-CH_2$ of rocking vibration of lipids (716 $cm^{-1}$) as well as $-CH_3$ symmetric bending at 1372 $cm^{-1}$. Some contribution to a broad $-OH$ stretch, probably due to iron oxyhydroxides, was observed. However, this peak was not as intense due to the presence of less oxyhydroxides or structurally bound water. Additional peaks detected in the IRB biofilm formed in the modified Postgate C media and not in the Postgate C media included the band at 1600 $cm^{-1}$ corresponding to the COO- symmetric stretch of lipids [44], the band corresponding to amino acid side chains at 1508 $cm^{-1}$, and a band observed at 1288 $cm^{-1}$ corresponding to $\alpha$ helix of amide III (30% C-N str, 30% N-H bending vibration, 10% O=C-N bending, 20% other) in proteins [44,49]. The most intense peak of the IR spectra, at 1180 $cm^{-1}$ and 1002 $cm^{-1}$, corresponded to the presence of carbohydrates and possibly LPS bacterial cell wall CO-P, P-O-P components in the IRB biofilm [42–44]. The stretching of the phosphate ions was also present, as indicated by peaks at 1246 $cm^{-1}$ (asymmetric $PO_2$ stretching) and 1076 $cm^{-1}$ ($PO_2$ symmetric stretching) and asymmetric P-O stretch at 826 $cm^{-1}$ [44]. The dwarf peaks at 696 $cm^{-1}$ and 674 $cm^{-1}$ assigned to C=O bending and $CH_2$ rocking vibrations, respectively [50,51].

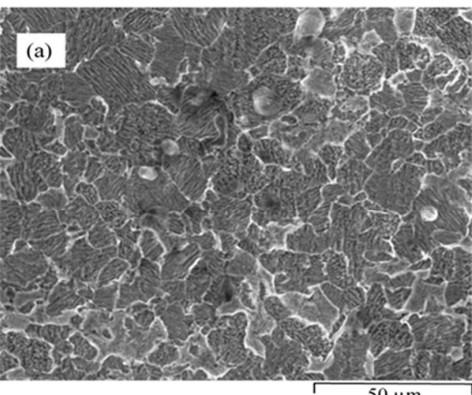
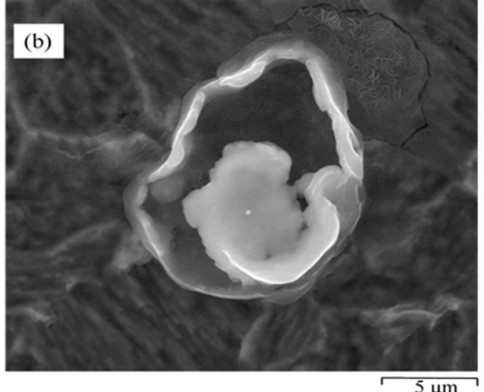

**Figure 7.** *Cont.*

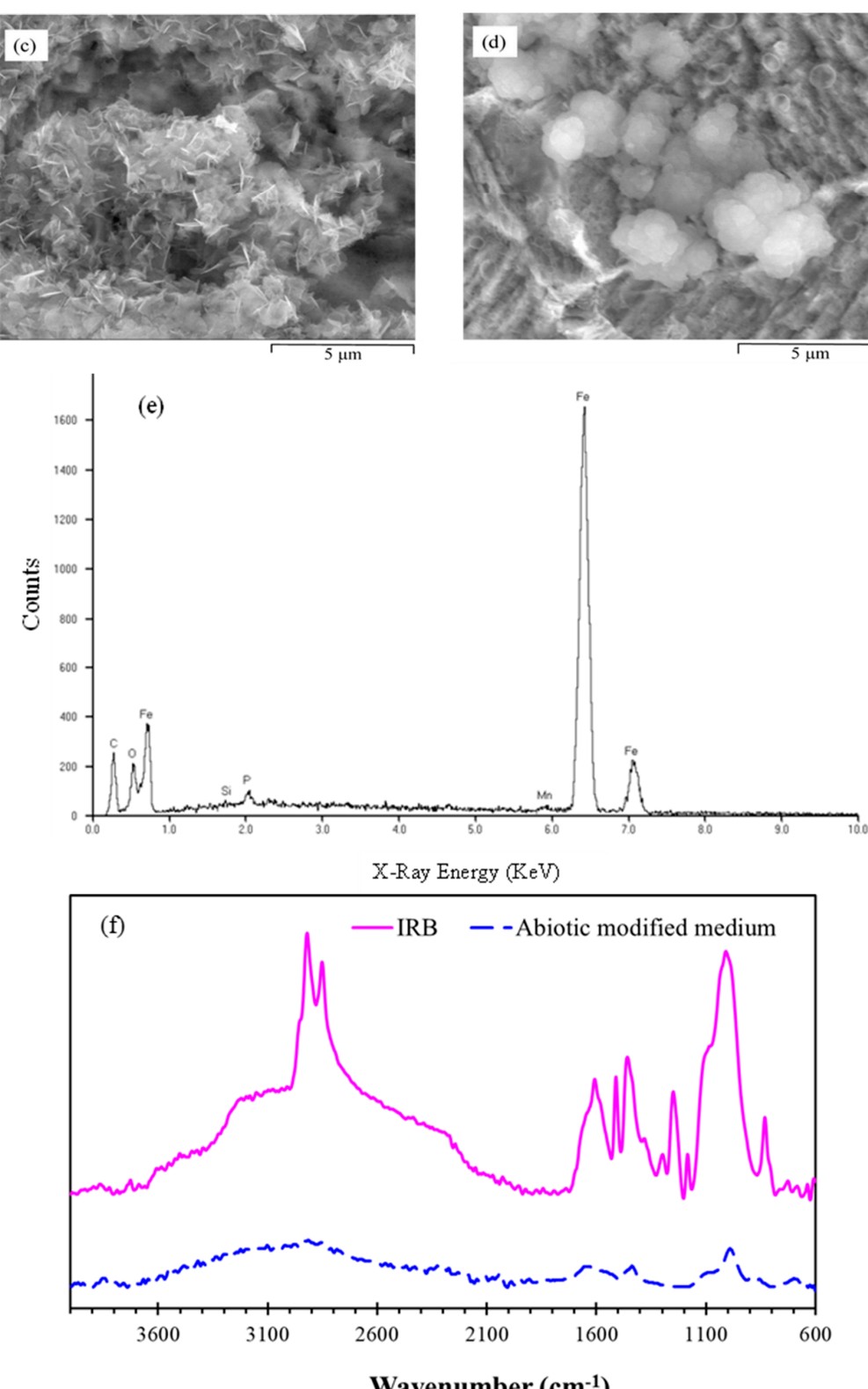

**Figure 7.** ESEM analysis of hydrated biofilm formed on carbon steel by a pure IRB culture following 72 h exposure to modified Postgate C medium: appearance of uniform corrosion (**a**) across the surface of carbon steel, a hollow iron oxide shell (**b**) at the surface of a corrosion pit, and needle (**c**) and globular (**d**) forms of reduced iron oxyhydroxide compound. (**e**) EDS spectra of the biofilm. (**f**) IR spectra of IRB biofilm formed on carbon steel exposed to pure IRB culture in modified Postgate C medium for a period of 72 h.

IR spectroscopy (Figures 3e and 7f) as well as microscopic observations (Figure 4a) of the pure IRB biofilm showed that organic material in the form of EPS was more prevalent in the presence of *S. putrefaciens*. The adhesive nature of the EPS secreted by the IRB may have a role in facilitating attachment of other bacterial species and/or nutrients to the biofilm as well as in consolidating the biofilm structure. The binding of metal ions by negatively charged EPS carbohydrates and proteins has also been previously reported [52].

FIB-SEM analysis showed the IRB biofilm to be extremely porous (Figure 8a,b). As a result, this film did not provide much protection and was expected to facilitate transport of chemical species between the electrolyte and the carbon steel surface. Therefore, the surface of the carbon steel under the biofilm was rough and corroded. At higher magnification (Figure 8c), an initial colloidal iron oxyhydroxide layer can be observed that appeared to be adhering to the corroding carbon steel with the aid from a sticky EPS-like material.

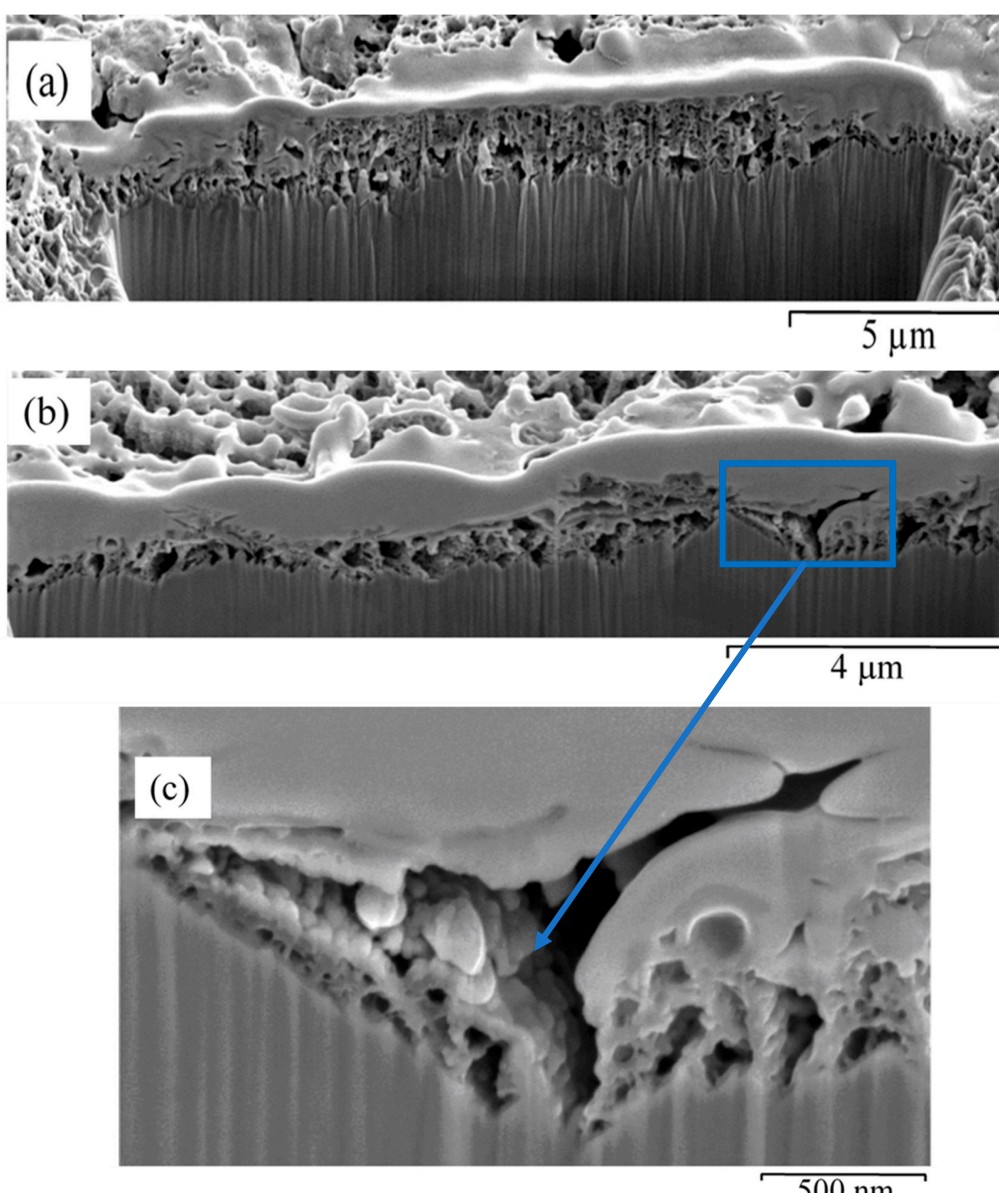

**Figure 8.** FIB-SEM images of the cross-sections (**a**,**b**) of biofilm formed two areas of a carbon steel surface by pure culture of IRB in modified Postgate C medium following a 72 h exposure a detailed view (**c**) of the corroded area of the cross-section identified in Figure 8b at higher magnification showing channels that could allow electrolyte transport.

The continuity of the IRB biofilm was observed to be of stochastic nature. In certain areas of the same sample, colloidal inorganic/organic precipitates of C, O, Na, P, S, Cl, Mn and Fe were observed (Figure 9a). Iron oxide shells covered in sticky EPS material, such as that shown in Figure 9b, were found scattered throughout the sample.

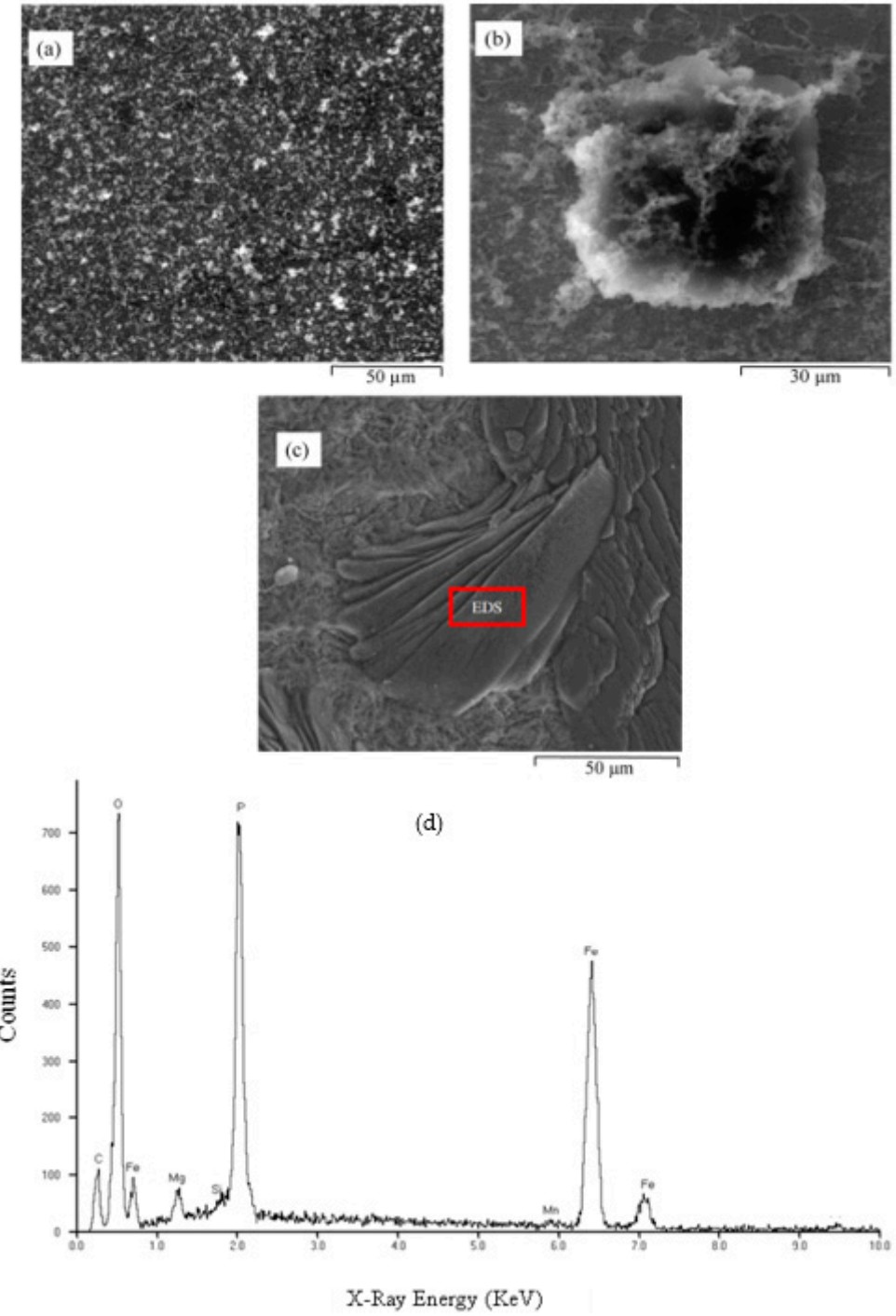

**Figure 9.** ESEM images of hydrated biofilm formed on carbon steel due to a pure IRB culture following 168 h exposure in the modified Postgate C medium: (**a**) overall covering (inorganic and organic components), (**b**) an iron oxyhydroxide shell covered in adhesive EPS material and (**c**) SEM image of precipitates of an iron phosphate species. (**d**) EDS analysis of the layered, plate-like deposit in Figure 9c.

The layered, plate-like deposit observed in Figure 9c was confirmed to be an iron phosphate species by EDS analysis as shown in Figure 9d. Glasauer et al., have reported that vivianite can form when sufficient $Fe^{2+}$ and phosphates are available [22]. As discussed earlier in Section 3.2.2, the morphology of these minerals was more easily observed under SEM than ESEM. It is likely that dehydration, which is most likely under SEM set up, changed the morphology of the minerals. FIB-SEM cross-section carried out under these iron phosphate deposits showed the total thickness of the layered mineral in Figure 9c to be approximately 10 µm. Pitting corrosion was not observed under iron phosphate deposits.

### 3.3.2. Electrochemical Behaviour of Steel in Modified Postgate C Medium with IRB

The build-up of biofilm was monitored using the OCP technique for a period of 168 h (Figure 10a). The negative shift in $E_{corr}$ that occurred in the presence of IRB biofilm due to either a decrease of cathodic reactant $O_2$ caused by IRB aerobic respiration and/or an increase in the anodic dissolution of iron caused by a loss of protective nature of the passive iron oxyhydroxide film. During the OCP measurement, $E_{corr}$ stabilized quickly and did not oscillate a great deal, suggesting low oxygen conditions in solution. $O_2$ consumption appeared to occur instantaneously as shown by a drop in $E_{corr}$ from initial reading of $-744$ mV at 0 h down to $-842$ mV at 72 h. The $E_{corr}$ then became stable during 72–168 h period where anaerobic or microaerobic conditions were set up by the bacteria. Consistent with this results, Sherar et al., reported an $E_{corr} < -800$ mV with anaerobic corrosion of carbon steel [53]. Once anaerobic conditions ensued, the corrosion of carbon steel was controlled by a combination of $H_2O$, $HCO_3^-$ and $Fe^{3+}$ reduction reactions.

Figure 10b depicts potentiodynamic polarization curves of carbon steel after 72 h exposure to the biotic and abiotic modified Postgate C solution. Active corrosion behaviour of carbon steel occurred in both the IRB and abiotic control. Corrosion resistance was similar in the IRB and abiotic control following 72 h pre-exposure (i.e., there was insignificant change in $E_{corr}$ and $I_{corr}$). However, following 168 h exposure, the $E_{corr}$ in the presence of the IRB was lower than that observed under abiotic conditions (Figure 10c), indicating the carbon steel to be more susceptible to corrosion in biotic solution without yeast after 168 h exposure. The corrosion current density of carbon steel in biotic solution was found to be about 9 times higher than that in the abiotic solution. These findings are consistent with the FIB-SEM observation, where pits are observed under the biofilm during the 168 h exposure to biotic Postgate C without yeast. Pits appeared to have propagated sideways and vertically. The slower increase in anodic current density in the potential range of $-600$ to $-800$ mV can possibly be attributed to the imbedding of corrosion products into the bacterial biofilm, thereby impeding the diffusion across the biofilm, which is consistent with the literature [9]. $E_{pit}$ after 168 h exposure to biotic medium is observed around $-575$ mV. At applied potentials more positive to $E_{pit}$, active corrosion similar to the abiotic control occurred (Figure 10c).

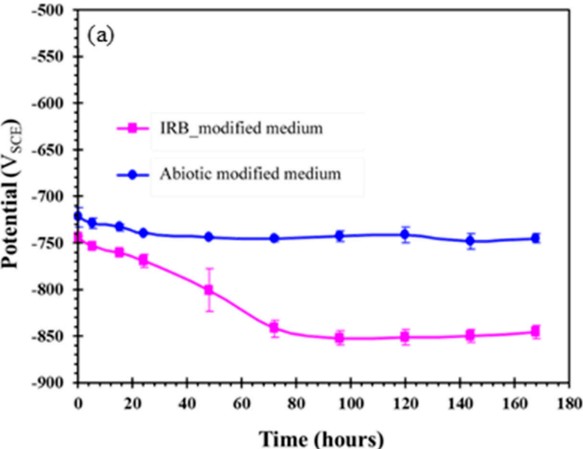

**Figure 10.** *Cont.*

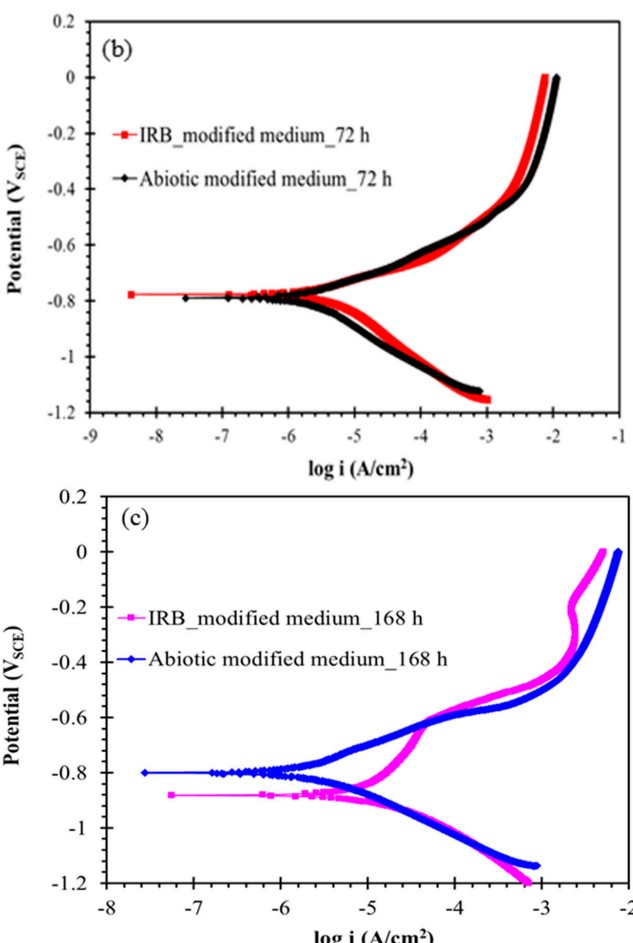

**Figure 10.** (**a**) Change in OCP of carbon steel with time of exposure to a modified Postgate C medium with pure IRB culture and without the culture (abiotic). (**b**) Potentiodynamic polarization scans of carbon steel after 72 h exposure in modified Postgate C medium with and without IRB. (**c**) Potentiodynamic polarization scans of carbon steel after 168 h exposure in modified Postgate C medium with and without IRB.

*3.4. Biofilm Development and Electrochemical Characterization of Carbon Steel in Biotic Postgate C Solution without Organic Nutrients (i.e., Inorganic Medium)*

3.4.1. Biofilm Development on Carbon Steel Exposed to IRB in Inorganic Postgate C Solution

In the absence of organic nutrients such as yeast extract, lactic acid and sodium citrate, the most likely electron donor for the IRB is hydrogen [54,55]. Not all IRB species can couple $H_2$ oxidation with growth [39,56]. However, Lovely et al. [39] have reported on the ability of *S. putrefaciens* to metabolize hydrogen at partial pressures 25-fold lower than that by the pure cultures of SRB. The hydrogen for bacterial metabolism would be generated by the reduction of water Equation (1). The reduction of water would also be the most likely cathodic reaction at neutral pH and limited $O_2$ environments for the corrosion of the carbon steel.

$$2H_2O + 2e^- \rightarrow H_2 + 2OH^- \tag{1}$$

The IRB may then utilise the $H_2$ generated through Equation (2) and couple with the reduction of $Fe^{3+}$ ions from ferric oxyhydroxides with the oxidation of $H_2$ Equation (2) [39];

$$H_2 + 2Fe^{3+} \rightarrow H^+ + 2Fe^{2+} \tag{2}$$

Previous studies have shown that IRB require direct contact with the Fe(III) minerals for bioreduction under nutritionally limiting conditions [21,37]. In contrast, other studies have observed that microbial siderophores (extracellular chelating agents) were involved in

reductive dissolution of ferric oxyhydroxides and oxides [57]. Glasauer et al. [22] observed that *S. putrefaciens* grown under nutrient-limited conditions only reduced poorly crystalline hydrous ferric oxides. Due to the absence of lactate in the inorganic medium, the formation of siderite, according to Equation (3), is not expected as the availability of bicarbonate in solution occurs due to oxidation of lactate coupled with iron oxyhydroxide reduction [25]. However, the precipitation of magnetite Equation (4), vivianite Equation (5) or green rust (GR) Equation (6) is still possible under different conditions of redox potential, available Fe(III) and/or Fe (II) and appropriate ligands.

$$Fe^{2+} + HCO_3^- \ \rightarrow FeCO_3 + H^+ \tag{3}$$

$$2Fe(OH)_3 + Fe^{2+} \rightarrow Fe_3O_4 + 2H_2O + 2H^+ \tag{4}$$

$$3Fe^{2+} + 2HPO_4^{2-} + 8H_2O \ \rightarrow Fe_3(PO_4)_2.8H_2O + 2H^+ \tag{5}$$

$$2Fe(OH)_3 + 4Fe^{2+} + A^{2-} \left( Cl^-, CO_3^{2-}, SO_4^{2-} \right) + 9H_2O \ \rightarrow GR + 6H^+ \tag{6}$$

In contrast to the IRB biofilms formed in complete Postgate C and the modified Postgate C mediums, there was lesser incidence of crystalline iron oxyhydroxides observed on the carbon steel surface when exposed to IRB in the inorganic medium. This may be due to the rate of crystalline iron oxyhydroxide formation being slower than the biotic iron reduction reaction [25]. Some colloidal particles observed in the IRB biofilm (Figure 11a) are speculated to be magnetite. EDX analysis shows the presence of Fe, C and O corresponding to the elemental composition of typical area, such as that shown in Figure 11a, indicating that iron phosphates may also be present as was also observed earlier under inorganic abiotic condition. The presence of iron oxyhydroxide shells in the biofilm became more recognisable in SEM analysis (Figure 11b) than from hydrated biofilms observed through ESEM. The shells in this case were observed to be broken down, either by the environmental conditions in the IRB solution or due to sample dehydration under SEM chamber. FIB cross-sectional analysis could not be performed with the intact hollow shells due to the fragile nature of the shells. However, as shown in Figure 12, FIB-SEM cross-sectional analysis could be performed on the broken iron oxyhydroxide shell (Figure 11b).

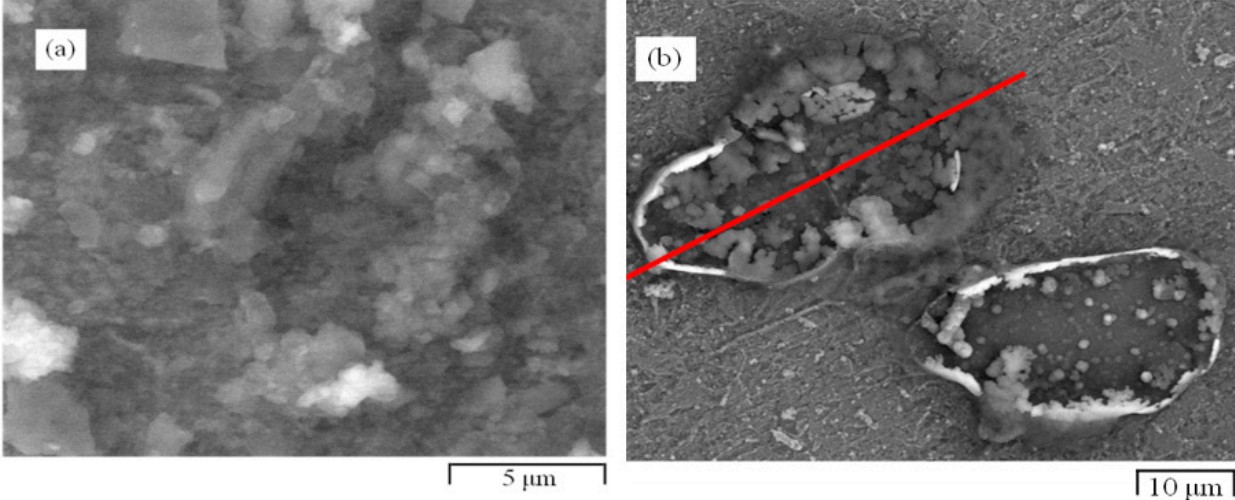

**Figure 11.** ESEM images of hydrated biofilm formed on carbon steel by a pure IRB culture following 72 h exposure to IRB and Postgate C without organic constituents: (**a**) amorphous corrosion product deposits containing carbon, oxygen phosphorous and iron. (**b**) SEM image of iron oxyhyrdroxide shell. FIB cross-section was carried out across the area indicated by the red line.

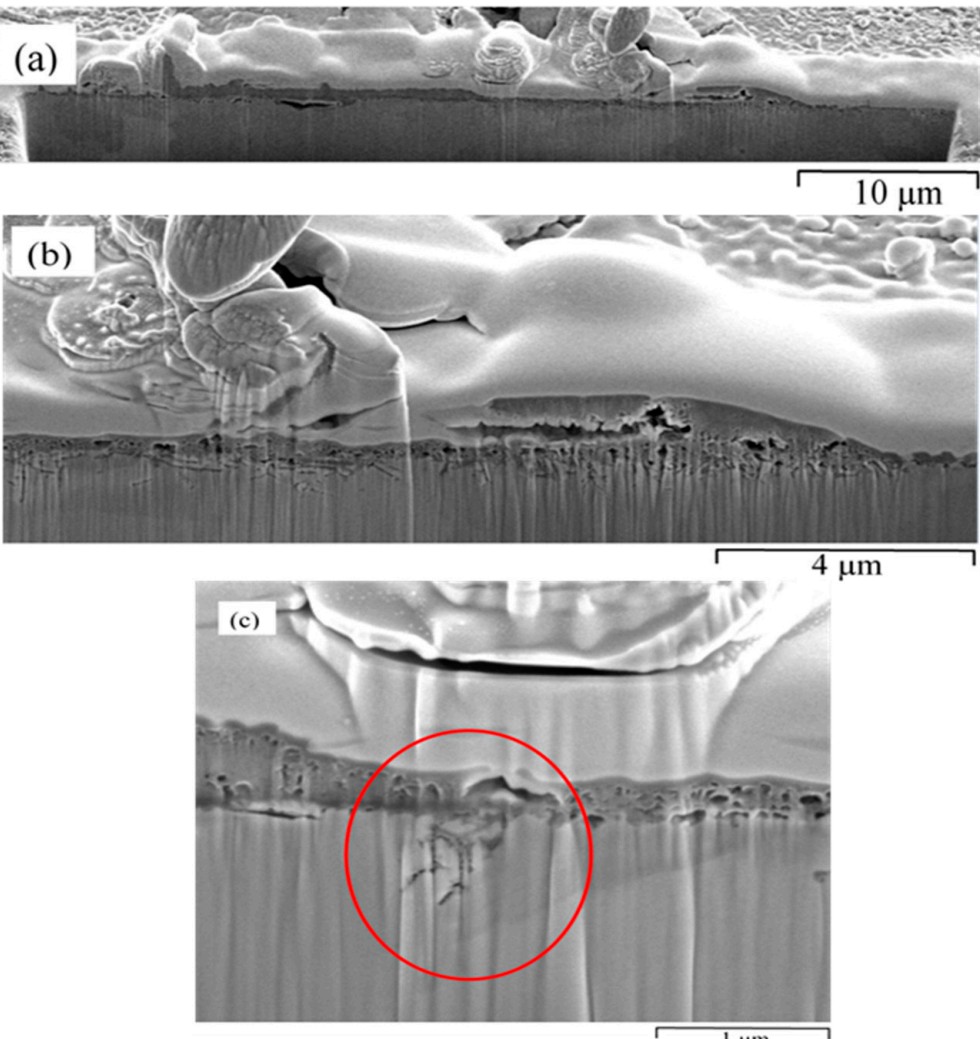

**Figure 12.** FIB-SEM cross-sectional analysis of biofilm formed on carbon steel by pure culture of IRB in inorganic medium following a 72 h exposure: (**a**) overview of the biofilm cross-section, and (**b,c**) the extent of cracking attack observed under the IRB biofilm at higher magnification.

The IRB biofilm formed under the inorganic environment was very thin. FIB-SEM cross-sectional analysis (Figure 12a) showed that the thickness was in the nm scale. When the cross-section was observed under higher magnification, extensive grooving into the steel surface was observed (Figure 12b). Figure 12c highlights an area of localized attack that progressed along susceptible points in the steel microstructure such as voids, inclusions or grain boundaries. Biofilm material was observed to have filled the grooves produced by the localized attack. There have been limited studies on the influence of IRB on corrosion, and this type of localized attack has not been previously reported in literature.

The localised attack shown in Figure 12c appears like a crack, which might occur because of $H^+$ cation build-up. $H^+$ is generated due to $H_2$ oxidizing ability of the IRB, coupled with ferric hydroxide reduction at localized areas. $H^+$ cations can transform into H atoms that may lead to the build-up of adsorbed hydrogen atom in the carbon steel subsurface. The H atoms may then combine to form $H_2$ molecules, particularly at the high energy locations such as voids and grain boundaries. The $H_2$ molecules are too big to diffuse out of the voids and the resulting build-up of gas pressure would cause cracking. Such cracking does not require externally applied tensile stress to progress [58]. Although initial signs of undercutting pitting attack were observed under the IRB biofilm in nutritionally rich conditions (Figure 5), such cracking was only observed for the carbon steel exposed to IRB in the inorganic medium. This may be due to the IRB oxidizing organic lactate in these

two nutritionally rich media removes 7 moles of H$^+$ for every lactate molecule oxidized, as per Equation (7), and controls the build-up of H$^+$ ions. While previous studies have not investigated the MIC behaviour under the IRB biofilm, the acidic H$_2$S formation in the presence of the SRB has been reported to lead to MIC. H$_2$S enhances the adsorption of hydrogen atoms into the metal by poisoning the recombination of H atoms on the metal surface [58]. A previous study has used hydrogen permeation experiments and has observed enhanced hydrogen permeation due to cultures of SRB isolated from oil field waters [59].

$$4Fe(OH)_3 + CH_3CHOHCOO^- + 7H^+ \rightarrow 4Fe^{2+} + CH_3COO^- + HCO_3^- + 10H_2O \quad (7)$$

After 168 h exposure of the carbon steel to the IRB in the inorganic medium, micro-pitting could be observed (Figure 13a). The colloidal precipitates, such as these shown in Figure 13b, were scatted throughout the sample surface, and were identified as an iron phosphates species based on EDS. The polishing marks visible alongside these micro-pits suggest a non-equal distribution of the corrosion rate. Low corrosion rate has occurred in certain areas of the sample (Figure 13c), whereas deep localized pitting attack was also observed, as discussed below.

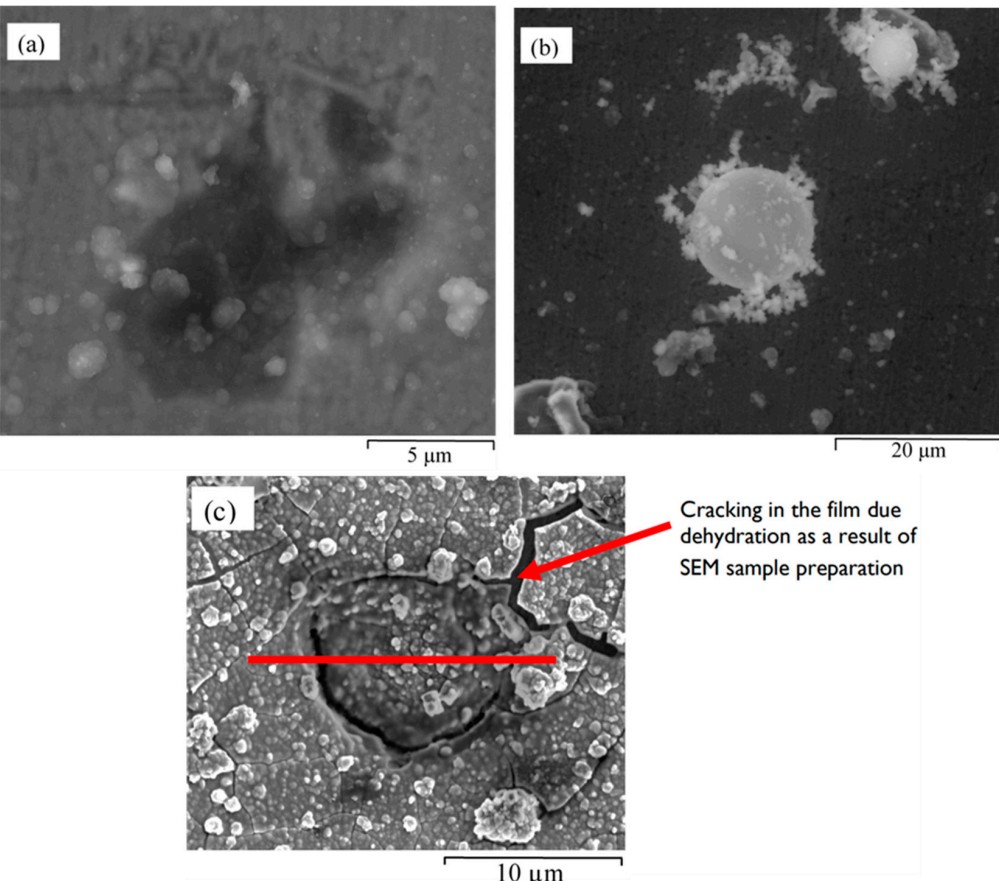

**Figure 13.** ESEM images of hydrated biofilm formed on carbon steel by a pure IRB culture following 168 h exposure in inorganic medium: (**a**) pits covered under corrosion deposit, (**b**) micro-pits in an area of low corrosion rate, and (**c**) SEM image showing the location where the cross-sectional FIB milling was performed.

EDS carried out at the centre of one of the dark patches observed in ESEM (Figure 13a) detected iron, oxygen and phosphorous to be the main elements. FIB-SEM cross-sectional analysis was carried out on such an area to investigate if these were sites of localized pitting attack. Figure 13c is a SEM image of the area and the red line is the location of FIB

cross-sectioning. Cracking of the film observed in the SEM images (Figure 13c) were not observed under ESEM for the same sample and hence, the conclusion that the cracks were a result of dehydration of IRB biofilm in the SEM chamber.

FIB-SEM in Figure 14a confirmed the existence of a deep pit under the area shown in Figure 13c. A pit (~10 µm deep) developed underneath the biofilm. In previous studies on stainless steels, growth of the pit may occur back towards the outer surface to where the pit originated [60,61]. The mechanism for such behaviour has been attributed to the passivation of the surface near the pit mouth due to concentration of metal ions being below the critical concentration for pit propagation ($C_{crit}$). However, in the pit adjacent to the passivated area, $Fe^{2+}$ ions are generated and the dissolution front progresses through the sidewall until the passive film is punctured [61]. EDS analysis from the area highlighted in Figure 14a showed a high intensity peak for carbon that can only be attributed to the high content of organic material in the biofilm, since the test medium of Postgate C only consists of inorganic constituents. Inorganic sulphate and chloride were also present. It is assumed that $SO_4^-$ (as indicated by S and O in EDS) and $Cl^-$ ions moved to the corroding site to maintain electrical neutrality by balancing out the positive $Fe^{2+}$ generated from the corroding carbon steel. They may also be incorporated into green rust minerals according to Equation (6). Another notable mineral present on the pit covering material was the hollow iron oxyhydroxides shells. These hollow shells were observed near the side walls (Figure 14b) as well as being incorporated between other corrosion product layers as outlined in Figure 14c.

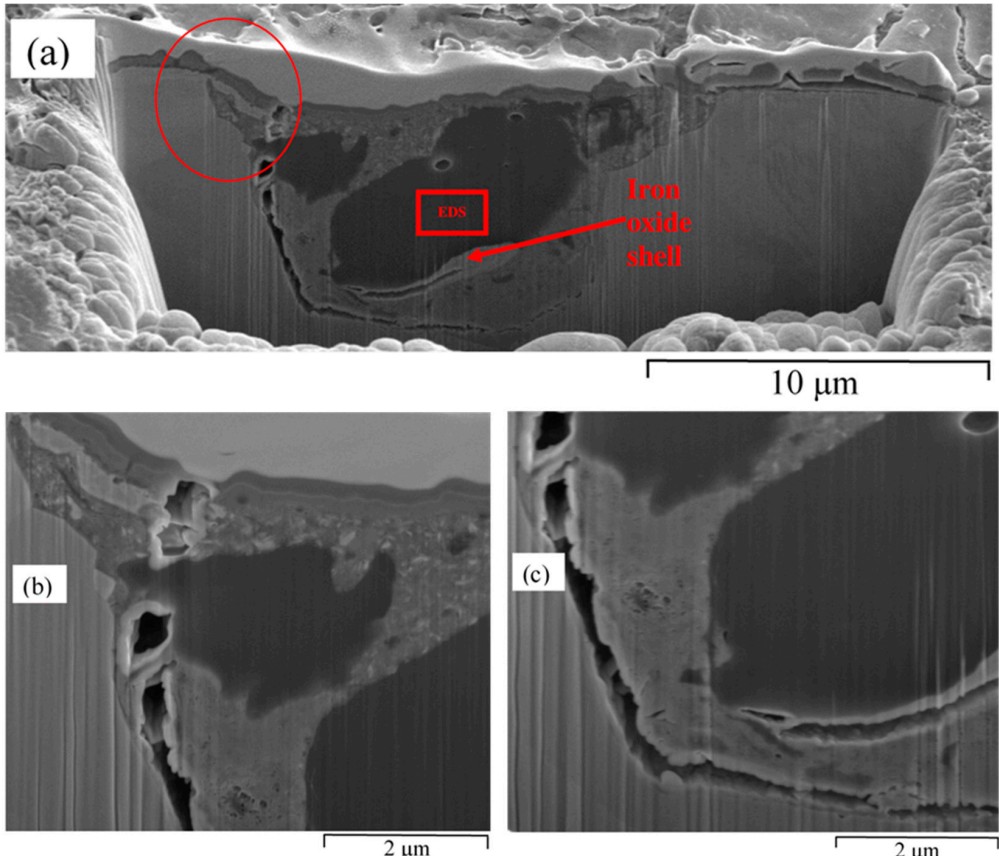

**Figure 14.** FIB-SEM cross-section of specific features of carbon steel exposed to the pure culture of IRB in the inorganic medium (i.e., Postgate C without organic components) for 168 h: (**a**) an overview of the cross-section (the circled area marks undercutting pitting attack making its way back towards the outer surface, EDS was performed in the area in the rectangle), (**b**) a close up view of the undercutting pit, and (**c**) a close up view of the side walls of the pit where biofilm is loosely attached in this area allowing electrolyte transport.

FIB-SEM cross-section characterization was carried out also for another location in the sample exposed for 168 h (Figure 15). In this case, the sample suffered some charging during SEM imaging, due probably to the high organic content in the IRB biofilm. Pit was covered under the corrosion product deposit (Figure 15b). The morphology showing an undercutting attack in Figure 15b is similar to pitting in acidic environments [62]. The cracking in the biofilm that was observed in the 72 h exposure to the inorganic medium occurred also in the 168 h exposure (Figure 15c). Biofilm extended into the cracks, as seen in the higher magnification images (Figure 15c). Similar to the 72 h exposure sample, EDS of the corrosion deposit covering the pit shown in Figure 15b detected a high intensity peak for carbon, indicating the presence of organic molecules. Other elements detected were sulphur, chloride and iron.

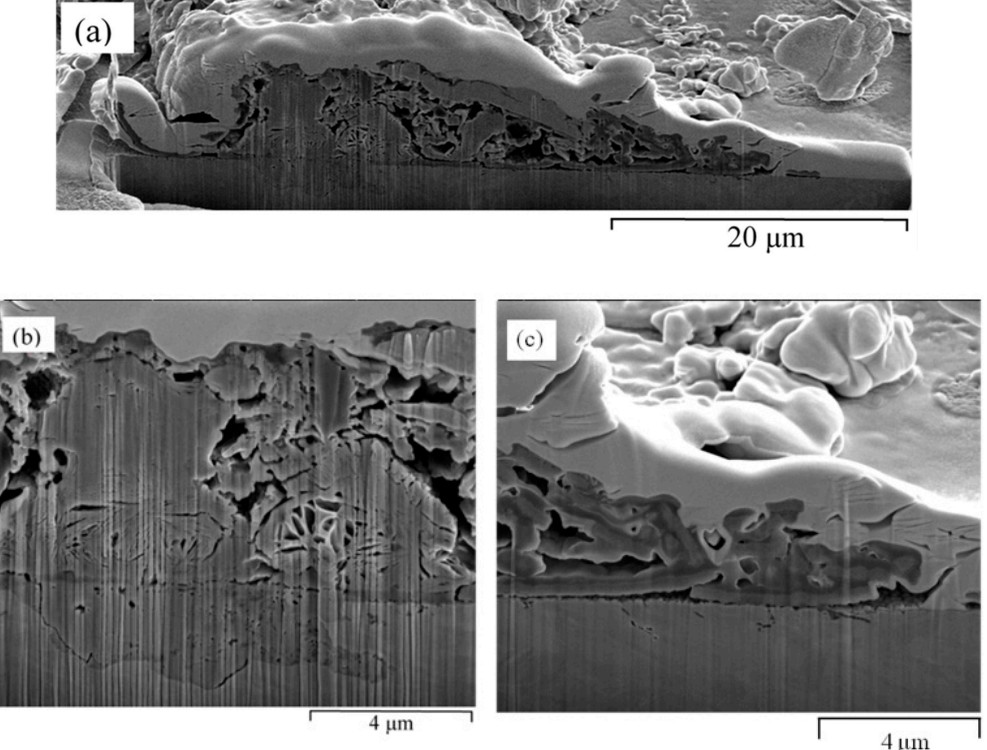

**Figure 15.** FIB-SEM cross-section of more common features of carbon steel exposed to the pure culture of IRB in the inorganic medium (i.e., Postgate C without organic components) for 168 h: (**a**) overview of a cross-section under an area of a corrosion deposit, (**b**) undercutting pitting has occurred under the biofilm, and (**c**) localised corrosion penetration and biofilm.

3.4.2. Electrochemical Behaviour Carbon Steel Exposed to Inorganic Postgate C Solution with IRB

Potentiodynamic polarization scans of carbon steel following 72 h exposure in the inorganic medium with and without the IRB are shown in Figure 16a. The corrosion potential of steel in inorganic biotic medium shifted by about 50 mV in a cathodic direction compared to that in the abiotic solution. The corrosion current density of the steel in biotic solution was at least an order of magnitude lower than that in the abiotic solution. As mentioned earlier, IRB are able to reduce the oxygen level, and limit the oxygen available for corrosion. It is also reported that in case of nutrient deficiency, IRB fail to survive; hence, there is no need of IRB forcing Fe(III) reduction to soluble Fe(II). Thus, the insoluble Fe(III) oxides (such as goethite) [63] persist and provide protection.

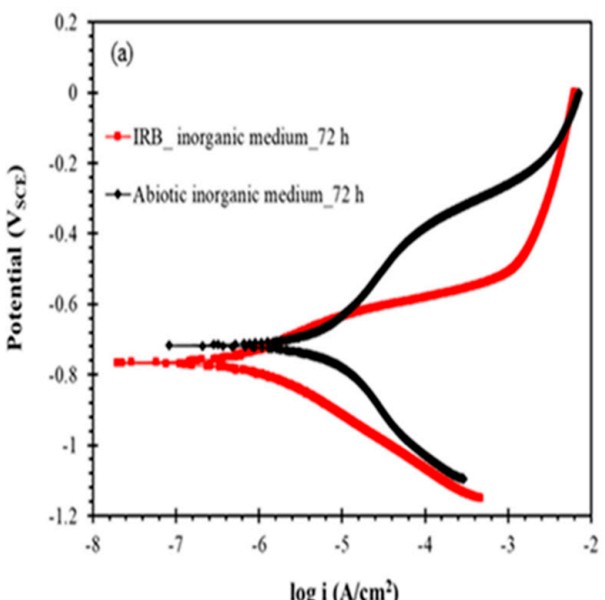 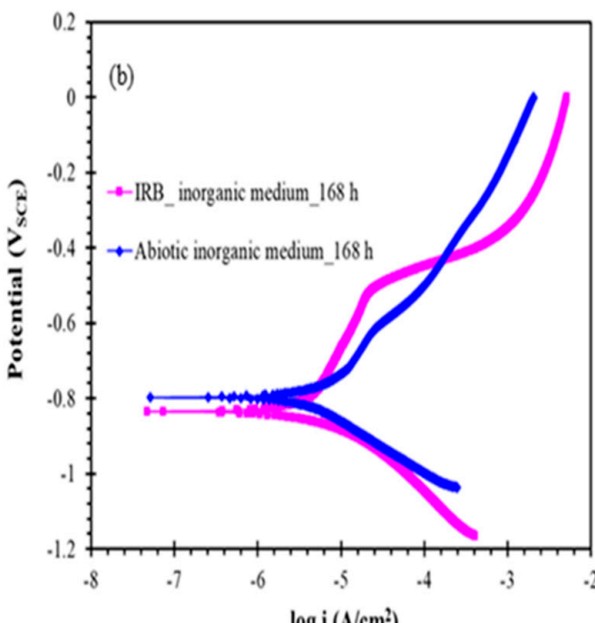

**Figure 16.** (**a**,**b**) Potentiodynamic polarization scans of carbon steel following 72 and 168 h exposure, with and without the IRB, to inorganic Postgate C solution.

Figure 16b shows the potentiodynamic polarization scan following 168 h exposure to IRB and the abiotic control inorganic Postgate C solutions. Though the corrosion resistance of steel decreased when the duration of exposure to inorganic IRB solution increased to 168 h, the corrosion current density of steel specimen in biotic solution still greater by >1.5 times than that of steel exposed to the abiotic solution. The sluggish increase in anodic current density at the anodic potential range of 0.74–0.58 $V_{SCE}$ could be attributed to the formation of the ferrous species and rust during the anodic polarization. $E_{pit}$ was observed at −550 mV, and as the applied potential was increased to more positive values from $E_{pit}$, active corrosion was observed.

No previous studies have investigated the influence of nutrients on the IRB biofilm formation or study the cross-section of the IRB biofilm. Results of this study show that the accelerated corrosion observed in IRB environments, when cultured under starvation conditions, requires further investigation.

## 4. Conclusions

*S. putrefaciens* was observed to be biofilm forming bacteria. The highest coverage to EPS material as well as reduced iron/oxyhydroxides were observed in the IRB biofilm in the presence of yeast extract in the Postgate C medium. The IRB contained a higher carbohydrate portion in the biofilm than the abiotic corrosion product film.

The build-up of organic material and reduced minerals did not provide any significant protection to the metal and active corrosion occurred when external potential was applied. Dissolution of Fe(III) oxyhydroxides occurred in the presence of the IRB which led to a greater dissolution of $Fe^{2+}$ ions into the solution than under abiotic conditions. The IRB were also able to set up anaerobic conditions in solution by the removal of residual $O_2$ present in solution.

Undercutting pitting corrosion morphology was observed in the presence of the IRB in the nutrient-rich media, and to a greater extent, in nutrient-limited conditions. Deep pits on carbon steel were observed when IRB were cultured in the inorganic medium. From the results of this study, it can be concluded that the IRB biofilm does not protect carbon steel from corrosion and was observed to accelerate abiotic corrosion processes in a localised manner.

**Author Contributions:** S.W.; Conceptualization, Data curation, Formal analysis, Investigation, Methodology, Software, Validation, Visualization, Writing—review & editing. S.A.-S.; Investigation, Methodology, Formal analysis, Validation, Writing—review & editing. W.P.G.; Conceptualization, Investigation, Methodology, Formal analysis, Validation, Writing—review & editing. R.K.S.R.; Conceptualization, Investigation, Methodology, Formal analysis, Validation, Writing—review & editing. C.P.; Investigation, Methodology, Writing—review & editing. All authors have read and agreed to the published version of the manuscript.

**Funding:** This research received no external funding.

**Informed Consent Statement:** Not applicable.

**Data Availability Statement:** Data is contained within the article.

**Acknowledgments:** Authors acknowledge the use of microscopic facilities at Monash Centre for Electron Microscopy (MCEM).

**Conflicts of Interest:** The authors declare no conflict of interest.

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
