# Peer review of "Biofilm Development on Carbon Steel by Iron Reducing Bacterium Shewanella putrefaciens and Their Role in Corrosion"

_metals, doi:10.3390/met12061005_

Round 1

Reviewer 1 Report

This work is to characterise the IRB biofilm on carbon steel, and its effect of on corrosion characteristics. Some results were obtained. The paper is interesting but need to address following points before it could be accepted for publication

1.  Please refine the Materials and Methods.

2.  Line 229 (specimen was covered by amorphous EPS-like material (Figure 3)), how do you confirm that amorphous?  

3.  Please mark the phosphate correspond to the peaks in IRB, such as Fig,3e.

4.  Line 326 “EDS analysis of the biofilm corresponding to the ESEM image”, Please show the result of EDS.

5.  L330-333” One such deposit was chosen for the FIB-SEM cross 330 sectional characterization (Figures 5c and 5d). At high magnification, the thin biofilm be- 331 side the deposit showed initial signs of undercutting attack as the metal was dissolved 332 from underneath.” How do you distinguish the biofilm and corrosion product?

6.  How do you get the Rp?

7.  L376 “the hollow iron oxyhydroxide 376 shell” how do you know it is iron oxyhydroxide?

8.  Please add the result in Fig.9

9.  The figure quality is poor such as Fig.11 and Fig.15, please replace them.

Author Response

Please the attached file.

Reviewer 2 Report

MIC is a kind of complicated corrosion process due to the activities of microorganisms are involved. Comprehensive efforts had been carried out in this work to characterise the IRB biofilm on carbon steel, and its effect of on corrosion characteristics, contributing to the understanding of the IRB influenced corrosion mechanisms. It is confirmed that S. putrefaciens is biofilm forming bacteria, the undercutting pitting corrosion morphology was more pronounced observed in the presence of the IRB in nutrient limited conditions than in the nutrient-rich media. It is cleared that the IRB biofilm does not protect carbon steel from corrosion bit to accelerate abiotic corrosion processes in a localised manner. There for I suggest minor revision after the following issues are considered if applicable.

  1. Line 18: ‘ATR-FTIR’, abbreviations should be used after they are defined when they appear first.
  2. Line 210: Fig.1, the photo of day 22 is suggested to be adjusted to be parallel with others. If clearer photos are available, better to be substituted.
  3. If electrochemical impedance spectrum (EIS) technique is available, more information might be derived. (Not compulsory. Maybe in future work)
  4. It is concluded that IRB accelerates abiotic corrosion processes in a localised manner. Can this kind of localized corrosion be characterized by cyclic voltammetry?

Reviewer 3 Report

This paper was well prepared and organized from the point that it showed the biofilm formation and its effect on corrosion of carbon steel. But some part of the manuscript needs to be revised as follows;

Line 18; ATR-FTIR needs the full name.

Line 106; Three recipes was newly formulated by your group in this work? If not, they need some references.

Line 137; [3  1] is [31] or [3, 1]? But please describe the test procedure in detail.

Line 138; 2.5 was too long and thus they needs the brief description.

Figure 1 needs the detail experimental condition including temperature etc.

Figure 2; Increasing test time, total iron concentration was decreased. Please explain it. The effects of IRB or Abiotic were different. Why?

Figure 3 & 7; Peaks needs the symbol in the figure(-CH3 or -CH2 etc.) Did you detect the chemical composition of biofilm in ESEM image ?

Figures on the cross section; Did you show whether the matrix of carbon steel was corroded or not?

Figure 6(a); What did you use the equivalent circuit? Please describe the method about AC impedance measurement in the experimental methods.

Every figures need the identification when the black and white printed.

Line 663; Conclusions need the quantitative effects of IRB and abiotic on corrosion of carbon steel, if possible.

Round 2

Reviewer 1 Report

The results are interesting, and it can be accepted.
